# Sensitivity of Papilledema as a Sign of Increased Intracranial Pressure

**DOI:** 10.3390/children10040723

**Published:** 2023-04-14

**Authors:** David Krahulik, Lumir Hrabalek, Filip Blazek, Matej Halaj, Marek Slachta, Eva Klaskova, Klara Maresova

**Affiliations:** 1Department of Neurosurgery, University Hospital Olomouc, 779 00 Olomouc, Czech Republic; 2Department of Pediatrics, University Hospital Olomouc, 779 00 Olomouc, Czech Republic; 3Department of Ophthalmology, University Hospital Olomouc, 779 00 Olomouc, Czech Republic

**Keywords:** papilledema, optic disc swelling, increased intracranial pressure

## Abstract

Our study evaluates the sensitivity of papilledema as a sign of high intracranial pressure in children. Patients younger than 18 years old, diagnosed with increased ICP, and who had received dilated fundus examination between 2019 and 2021 were retrospectively reviewed. Factors including the patient’s age, sex, aetiology, duration of signs or symptoms, intracranial pressure (ICP), and presence of papilledema were evaluated. We included 39 patients in this study, whose mean age was 6.7 years. The 31 patients without papilledema had a mean age of 5.7 years, and 8 patients (20%) with papilledema had a mean age of 10.4 (*p* < 0.037). The mean duration of signs or symptoms was nine weeks in patients without papilledema and seven weeks in those with papilledema (*p* = 0.410). The leading causes of increased ICP with papilledema were supratentorial tumor (12.5%), infratentorial tumor (33.3%), and hydrocephalus (20%) (*p* = 0.479). Papilledema was statistically significantly more common in older patients. We found no statistical significance between sex, diagnosis, and symptoms. The relatively low incidence of papilledema (20%) in our study shows that papilledema’s absence does not ensure the absence of increased ICP, especially in younger patients.

## 1. Introduction

The optic nerve develops as an outgrowth of the diencephalon. Therefore, like the rest of the central nervous system (CNS), it is protected by derivatives of the brain meninges and washed by cerebrospinal fluid in the subarachnoid space. This arrangement differs from other cranial nerves.

Due to this morphology, we can observe an increase in intracranial pressure (ICP) in the optic nerve, manifested as papillary swelling. The pathogenesis of papilledema with elevated ICP has been elucidated in many studies in the past [1]. Elevated pressure is relayed from the subarachnoid space through the nerve to the papilla. This pressure disrupts the flow of the axoplasm, leading to its accumulation inside the nerve fibre and subsequent swelling [1]. Visual oedema of the optic nerve also compresses vessels with a thin and compliant wall (capillaries and venules) in the prelaminar region. However, vascular changes are only a secondary phenomenon. It does not appear until the oedema of the optic disc manifests itself [2].

The brain is capable of autoregulating cerebral circulation. This means that blood flow from the carotid and vertebral arteries is regulated to constant values, preventing pressure fluctuations. Autoregulation is determined by the relationship between cerebral perfusion and cerebral perfusion pressure.

The physiological value of ICP changes during spontaneous respiratory activity. For children, the limit is up to 20 cmH_2_O [3]; in new-borns up to 15 cmH_2_O. 

According to the Monroe–Kellie doctrine, the increase in intracranial pressure can occur by enlarging one of the three compartments: brain tissue, cerebrospinal fluid, and blood.

Clinically, ICP elevation is manifested by headache, nausea, and vomiting. In young children with hydrocephalus, the head circumference can enlarge [4,5].

Based on the mechanisms described above, papilledema thus might become an essential diagnostic element. Left unnoticed, it can lead to permanent sight damage (optic atrophy) [5]. However, as many previous studies have shown, optic nerve papillary eye examinations are not specific or sensitive enough to assess the presence or development of ICP value [2,4,5,6].

Steffen’s et al. (1996) study shows that papilledema is not an indicator of acute ICP elevation. They also point out that it depends on the duration of increased ICP rather than on its sharp rise [1]. Other studies from which Steffen et al. indicate that evidence of papilledema with increased ICP in cranial trauma and intracranial haemorrhage is relatively rare ^1^. Levatin and Raskind (1973) further found that papilledema may continue to develop despite therapeutic management of the acute phase of intracranial hypertension and adjustment of ICP [2,7].

However, papilledema appears to be a diagnostic guide in hydrocephalus. In his study, Lee shows that a higher percentage of papilledema was observed in patients with hydrocephalus due to a brain tumor [5]. However, the absence of papilledema is no reason to rule out intracranial hypertension [5]. The absence of papilledema was observed in up to 41% of children with hydrocephalus and would lead to false negatives in the diagnosis of hydrocephalus [5]. Allen et al. (1993) confirm in their study that papilledema was absent in 35% of children with a primary brain tumor [1]. Further research was conducted and it showed that children without detected papilledema were younger than children with proven oedema [5]. This phenomenon is explained by open fontanelles in early childhood.

Moreover, if the children have no additional signs or symptoms, with normal ICP and no vision loss, comprehensive non-invasive testing, e.g., ultrasonography, will distinguish abnormal optic nerve from pseudopapilledema. The main difference between true disc oedema and pseudopapilledema is the oedema of nerve fibre layer, which is present in the true form. It is crucial for ophthalmologists to differentiate between these two as incorrect diagnoses can lead to false positives.

## 2. Materials and Methods

Our single centre retrospective study included all paediatric patients (younger than 18 years of age) between July 2019 and April 2021, admitted to our centre in Olomouc, Czech Republic, who met one of the following criteria: ICP > 20 cmH_2_O due to hydrocephalus (1), original shunt failure (2) or newly diagnosed intracranial tumor (3). Children with an initial eye disorder unrelated to hydrocephalus or failure of the original shunt were excluded from this group. These patients were excluded to ensure adequate fundoscopic examination by an ophthalmologist. Our study aimed to determine the proportion of patients with increased ICP with concurrent papilledema as well as papilledema’s reliability as a marker of increased ICP.

The entry criteria for elevated ICP were set as follows: shunt malfunction was confirmed perioperatively, and a neurosurgeon measured the ICP. ICP in patients with intracranial tumors were measured using a ventricular catheter or ICP sensor. All patients also underwent fundus examinations by the same ophthalmologist with more than 15 years of experience for maximal consistency in the evaluation and differencing and excluding patients with pseudopapilledema. Papilledema and pallor were assessed in all our patients. Patients who did not meet these criteria were excluded from the study.

The finding on the optic disc was considered physiological by the ophthalmologist if it appeared “flat and pink”. Optic discs in which either elevation, retinal vascular darkening, cotton wool spots, or haemorrhage were present were evaluated as papilledema.

If the finding could not be evaluated reliably enough, such as an optic disc with severe pallor, which may not be able to develop papilledema even with ICP elevation, the patients were excluded from our study.

The following patient data were recorded for evaluation: age, sex, aetiology of hydrocephalus, the reason for shunt failure if already implanted, and manifesting symptomatology.

For all patients in the study, an imaging method was performed to assess the change in the conditions of the brain’s ventricular system. We set an ICP value > 20 cmH_2_O as cut off for intracranial hypertension. At the same time, we tried to correlate papilledema with age and symptoms according to their duration: headache or vomiting.

Thirty-nine children were included in the study. Statistical analysis was performed using IBM SPSS Statistics version 23 (Armonk, NY, USA, IBM Corp.). All tests were carried out at the 0.05 statistical significance value. We used the ‘Fisher’s exact test *p*-value’ where applicable, as it gives us and exact randomization distribution, which proves most useful in small samples, such as was in our group. For values with normal distribution, we used the ‘Mann–Whitney U Test *p*-value’ as a safeguard from wrong conclusions. For three different groups with recorded values, we used the ‘Kruskal–Wallis Test’. All the used tests were chosen to best fit the tested group variables to come to the most correct conclusions.

## 3. Results

All of the patient data has been split into several corresponding groups (Table 1, Table 2, Table 3, Table 4, Table 5 and Table 6) in which they are best compared in accordance with the clinical symptoms with which the patients might be admitted.

Our patient group included children across all age categories, with the youngest being 1 month old and the oldest 18 years, the average age of our patients was 6.7 years (Table 1).

Most of the patients were presented with vomiting and headache to name a couple from a variety, some of them were firstly referred to our centre after papilledema was diagnosed—Table 2. This table also shows the distribution of sex and diagnosis type. Headache in younger children has been evaluated by clinical signs when the child was first admitted to the hospital.

In Table 3, Table 4 and Table 5 we compare the presence of papilledema in accordance with the primary diagnosis, symptoms, and the duration of the symptoms, respectively. No statistically significant relevance has been found in our relatively small patient group, it is however essential to point out that the duration of symptoms does not increase the probability of papilledema forming.

The incidence of papilledema was statistically significantly higher in older children (*p* = 0.037) (Table 6 and Figure 1).

We compare the ICP values recorded in accordance with the primary diagnosis in Table 7, where statistically significant relevance has been found. The ICP values recorded were higher in patients with a diagnosis of infratentorial tumor.

Moreover, the average recorded ICP value for patients with confirmed papilledema was 23.75 cmH_2_O, irrespective of the primary diagnosis.

## 4. Discussion

The Canadian Preoperative Prediction Rule for Hydrocephalus (CPPRH) considers the presence of papilledema as a predictor of the development of persistent hydrocephalus in a posterior fossa tumor requiring shunt implantation. If the papilledema is undetected, it can lead to irreversible damage, such as optic nerve atrophy. However, the absence of papilledema could be a false-negative sign, and we try to correlate ICP and papilledema. In our study, papilledema was found only in 8 children (20%) with increased ICP. Another study, [1,5], reported that papilledema was absent in 35% of children with hydrocephalus, more precisely, in 41% of children where hydrocephalus was due to a primary brain tumor. According to our data, older children have a higher incidence of papilledema. One possible explanation for this phenomenon could be the presence of open fontanelles in younger children, which acts as a protective mechanism against ICP growth. The open fontanelle allows the child’s head to expand, so there is no further elevation of the ICP during the pathological process. Lee had previously stated in his study that his nine paediatric patients with increased intracranial pressure with open fontanelle had a normal ocular background without signs of papilledema [1,4,5,8]. Further observations led Havreh to the conclusion that papilledema does not appear until the fontanelle is closed, and thus the bones of a child’s skull are firmly attached [9,10]. All study authors warn clinicians not to correlate the absence of papilledema with low ICP [4,11,12,13,14].

On the other hand, papilledema appears to be a handy indicator of preoperative diagnosis of pathological ICP values. If papilledema is present in symptomatic paediatric patients due to ICP elevation-such as those displaying nausea, vomiting, or headache, it should be a clear signal for subsequent neurological examination or radiological imaging [15]. Other data show that if a posterior fossa tumor is detected in younger patients with a physiological papillary finding because of an open fontanelle, they are more likely to develop persistent hydrocephalus. For these reasons, regular follow-up checks on children are essential [6]. In our experience, if we diagnose a brain tumor in a child who already has closed fontanelles, the obstruction of the ventricular system occurs more often. This manifests itself, especially in the vicinity of the brainstem, by restricting or blocking the flow in the third chamber or Sylvius aqueduct, which leads to the development of obstructive hydrocephalus.

We have proven that papilledema as well as higher ICP are more commonly present in patient with infratentorial tumors. Given that brain tumors are the second most common solid tumors in children, 70% of which have an infratentorial location, this could be a helpful marker in the differential diagnosis [6,16].

It is known from previous studies that the time taken for ICP to develop is the fundamental factor in the development of papilledema [2,4,7,17,18]. This phenomenon is also supported by studies that have reported a low incidence of papilledema in patients with acute ICP elevation [2,7,9,17,19]. In Lee’s study of patients without evidence of papilledema, hydrocephalus factors associated with haemorrhage or infection outweighed the others [5,20,21,22,23]. In addition, there was no significant difference in the duration of symptoms in any of the study groups-with or without papilledema.

The authors are aware of the limits of this study. This retrospective study included only patients with an ophthalmologist examination before the neurosurgical intervention. All other patients who were not initially examined were excluded. The absence of papillary examination in these patients occurred due to a lack of time before surgery. Another group of patients excluded from the study was those who did not have an initial ICP value. Due to the careful selection of a small sample of patients, a selection bias may occur. The second limit of the study is the subjective evaluation of papilledema. On this basis, several patients with borderline findings may miss the correct diagnosis.

Based on the above, further research involving a more significant number of patients is needed to elucidate the sensitivity of papilledema.

## 5. Conclusions

In summary, our study has shown that the incidence of papilledema was only in 20% of patients; however, older children have a significantly higher incidence of papilledema (*p* = 0.037). We found no statistically significant correlation between papilledema and the diagnoses with increased ICP nor between vomiting, headage, the duration of these symptoms and papilledema. Papilledema should be interpreted as one of the definitive signs of ICP elevation. However, the absence of papilledema cannot preclude an increase in intracranial pressure during hydrocephalus or brain tumor development.

Increased ICP occurs more commonly in patients with infratentorial tumor growth and helps to point the clinician towards the correct diagnosis.

To conclude, a comprehensive insight into the patient’s clinical condition, appropriate radiological imaging methods, and eye examination are required to make the correct diagnosis.

## Figures and Tables

**Figure 1 children-10-00723-f001:**
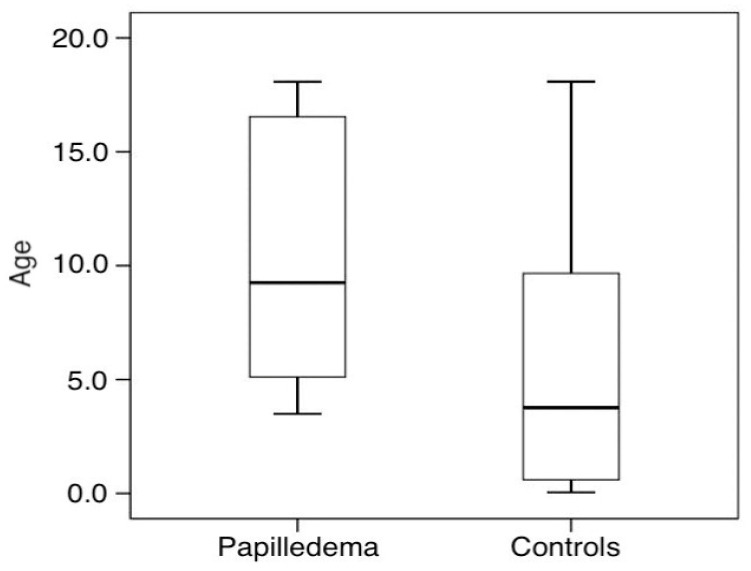
Box plot graph showing statistically significant correlation between papilledema and age.

**Table 1 children-10-00723-t001:** Age distribution in years in the group.

	Average	SD	Median	Minimum	Maximum
Age	6.7	6.2	4.8	0.05	18.1

**Table 2 children-10-00723-t002:** Distribution of the symptoms, sex, and diagnosis type in the following groups.

		Number	Percentage
Sex	M	21	53.8%
	F	18	46.2%
Diagnosis	TU supra	16	43.2%
	TU infra	6	16.2%
	Hydrocephalus	15	40.5%
Vomiting	Yes	13	35.1%
	No	24	64.9%
Headache	Yes	13	45.9%
	No	20	54.1%
Papilledema	Yes	8	20.5%
	No	31	79.5%

**Table 3 children-10-00723-t003:** Comparison of the presence of papilledema according to the diagnosis.

	Papilledema				Fisher’s Exact Test *p*-Value
	Yes		No		
	Number	Percentage	Number	Percentage	
TU supra	2	12.5%	14	87.5%	0.479
TU infra	2	33.3%	4	66.7%	
Hydrocephalus	3	20.0%	12	80.0%	

**Table 4 children-10-00723-t004:** Comparison of the presence of papilledema according to the symptoms.

		Papilledema				Fisher’s Exact Test *p*-Value
		Yes		No		
		Number	Percentage	Number	Percentage	
Vomiting	Yes	4	30.8%	9	69.2%	0.413
	No	4	30.8%	20	83.3%	
Headache	Yes	5	29.4%	23	70.6%	0.428
	No	3	15.0%	17	85.0%	

**Table 5 children-10-00723-t005:** Comparison of the presence of papilledema according to the duration of the symptoms in weeks.

		Average	SD	Median	Minimum	Maximum	Mann-Whitney U Test *p*-Value
Papilledema	Yes	7	6	5	1	37	0.410
	No	9	6	7	1	30	

**Table 6 children-10-00723-t006:** Comparison of the presence of papilledema according to the age.

		Average	SD	Median	Minimum	Maximum	Mann-Whitney U Test *p*-Value
Papilledema	Yes	10.4	6	9.3	3.5	18.1	0.037
	No	5.7	6	3.8	0.05	18.1	

**Table 7 children-10-00723-t007:** Comparison of ICP values according to the primary diagnosis.

	Average	SD	Median	Minimum	Maximum	Kruskal-Wallis Test *p*-Value
TU supra	23.13	2.45	23	20	28	0.00053
TU infra	28.83	1.47	28.5	27	31	
Hydrocephalus	21.53	1.37	21	20	24	

## Data Availability

The datasets used and/or analyzed during the current study are available from the corresponding author upon reasonable request.

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
