# Peer review of "Sensitivity of Papilledema as a Sign of Increased Intracranial Pressure"

_children, 2023, doi:10.3390/children10040723_

Round 1
Reviewer 1 Report
This retrospective study aims to study the correlation between papilledema and intracranial pressure in a pediatric neurosurgery population with shunt failure and tumor presentation. The topic of the study – the sensitivity and specificity of papilledema for increased intracranial pressure – is a worthy one. I do have some concerns about the methodology.
The definitions and inclusion criteria were inadequate. It is not clear how the patients were recruited – from a clinical database? At how many centers? Where in the world? It was not clear why 20cm of water was chosen as the limit for increased ICP. It was not clear who was diagnosing the papilledema (it said “same clinician” but was that the same clinician as the neurosurgeon, or a separate ophthalmologist?), nor how it was diagnosed, “flat and pink” are not usual diagnostic criteria. The grade of papilledema was also not specified. The statistical techniques were not explained or justified.
If the goal of the study was to understand the sensitivity and specificity of papilledema for raised ICP, then a different statistical approach could be considered. Rather than a series of tables with chi square tests, the authors could consider all of the variables together, with the outcome of ICP and the exposure of papilledema, controlling for age, duration of symptoms, etc. This kind of multivariable analysis would be much more illuminating and would allow the authors to arrive at sensitivities and specificities are both raw and adjusted.
Author Response
We had the manuscript reviewed by a native English speaker before resubmitting.
Reviewer 1
This retrospective study aims to study the correlation between papilledema and intracranial pressure in a pediatric neurosurgery population with shunt failure and tumor presentation. The topic of the study – the sensitivity and specificity of papilledema for increased intracranial pressure – is a worthy one. I do have some concerns about the methodology.
Thank you for your kind words about the importance of this study. We have addressed and incorporated your concerns in the comments below.
The definitions and inclusion criteria were inadequate. It is not clear how the patients were recruited – from a clinical database? At how many centers? Where in the world? It was not clear why 20cm of water was chosen as the limit for increased ICP. It was not clear who was diagnosing the papilledema (it said “same clinician” but was that the same clinician as the neurosurgeon, or a separate ophthalmologist?), nor how it was diagnosed, “flat and pink” are not usual diagnostic criteria. The grade of papilledema was also not specified. The statistical techniques were not explained or justified.
We thank the reviewer for this extensive comment. This was a single-centre retrospective study, where the patients were included from our internal clinical database in Olomouc, Czech Republic – we have now clarified that in the article. ICP over 20cmH2O was chosen as the limit as this is the cutoff value for intracranial hypertension at our centre.
The diagnosis of papilledema was done by the same ophthalmologist, as mentioned on the lines 83 and 88.
The pathologies the ophthalmologist looked for are mentioned on lines 92 and 93, ‘pink and flat’ were considered physiological – healthy.
If the goal of the study was to understand the sensitivity and specificity of papilledema for raised ICP, then a different statistical approach could be considered. Rather than a series of tables with chi square tests, the authors could consider all of the variables together, with the outcome of ICP and the exposure of papilledema, controlling for age, duration of symptoms, etc. This kind of multivariable analysis would be much more illuminating and would allow the authors to arrive at sensitivities and specificities are both raw and adjusted.
Reviewer 2 Report
My biggest concern with this manuscript is the lack of ICP data. If all patients had ICP measured why are these ICP measurements not included? I think it would be much more impactful data if these ICP measurements were included. Moreover, the methods would imply that any patient with a measured ICP > 20 cmH2O and a preop fundoscopic exam would be included but there is no mention of the ICP being sustained > 20 cmH2O for 5 minutes or more of monitoring. (We all know that persons without hydrocephalus can generate transient pressures in excess of 50 cmH2O with a forced Valsalva so some mention of the ICP being sustained > 20 cmH2O for 5 min or more for those patients with preoperative EVDs/ICP monitors is necessary. (I do think it is reasonable to infer that those shunt patients who had ICP measured under anesthesia at time of shunt revision were living at a pressure close to the pressure that they had measured in the OR for the duration of the their shunt malfunction symptoms as I am sure 5+ minutes of pressure recordings were not obtained for these patients. But, for this unique patient population in the study it would be valuable to know how far their measured ICP in the OR was from their baseline ICP or shunt valve pressure setting. We all know that ICP will go up with a shunt malfunction in a shunt dependent patient but how high it goes, or how much the pressure deviates from the patient's baseline pressure with a working shunt, can be quite different from patient to patient.) In the conclusion (line 142) it is mentioned that this paper is correlating ICP and papilledema but, without ICP measurements included I don't feel that this paper is in fact correlating these measurements (as correlating these measurements would, in my mind, necessitate providing both measured ICP values and papilledema grades 1-5 for each patient included in analysis). While not as critical as providing the ICP measurements, grading the papilledema observed would have been nice.
General comments:
- Paragraphs 3-5 (lines 44-57) seem unnecessary to the focus of the paper. While autoregulation, ICP changes with the respiratory cycle, and the Monroe-Kellie doctrine are are generally related to the topic of increased ICP these topic do not specifically have bearing on the focus of this paper: the relationship between ICP and the observation of papilledema.
- Why discuss pressure in torr in line 51 of the intro when it is discussed in cm H2O in the remainder of the paper?
- The last paragraph of the intro (lines 82-84) feels like a bit of a non sequitur. Maybe something that can be saved for a discussion section if you want to discuss CPPRH more. But, as the last statement in the intro, particularly as a stand alone paragraph, it doesn't make much sense.
- The discussion of IBM SPSS use and p valve of 0.05 (lines 114-116) should be included in the Methods section not Results
- You mention the significance of open versus closed fontanelles a lot but at no point in the paper do you tell the authors how many patients in your study had an open fontanelle. We can infer that at least one did (minimum age = 0.05) but given the median age of 4.8 I am guessing there were not too many patients with open fontanelles. This needs to be clearly stated if it is going to be such a focus in the Discussion section.
- While I agree that an open fontanelle is protective against fulminant global increases in ICP (as head enlargement and fontanelle bulging can help to prevent extreme increases in ICP), I not aware of data to support the statement in lines 164-165: "If we diagnose a brain tumor in a child who already has closed fontanelles, the obstruction of the ventricular system occurs more often." I have cared for many young patients with open fontanelles that develop tumor related obstructive hydrocephalus...while their hydrocephalus symptoms may be less acute/severe than slightly older children with closed fontanelles I don't know of data to support that young children are less prone to developing obstructive hydrocephalus. This would need references to support this statement.
- You mention the potential for selection bias in the discussion but I think this needs even more attention. In my pediatric neurosurgery practice I would generally not request a preoperative fundoscopic examination in a patient with clear clinical/imaging evidence of hydrocephalus/increased ICP. I only ask for an urgent preoperative fundoscopic exam in patients that have questionable signs of increased ICP on exam/imaging. If this is the case at your institution then this would constitute a massive selection bias far greater than what is described in the discussion. If your institutional standard is to make an effort to obtain a preoperative fundoscopic exam on all patients with possible (or definite) increased ICP then this should be stated clearly as this will help to allay some concerns with respect to selection bias.
- Lines 106-110: These two sentences don't go well together given that the imaging studies don't provide ICP values. I understand that ICP values were also recorded but having these sentences follow each other without a more detailed description of the ICP value measurements is confusing. Moreover, I'm a bit confused - wouldn't the tumor patients have also undergone preoperative imaging prior to going to OR or having an EVD or ICP monitor placed? Why are the shunt patients being singled out with respect to the mention of preoperative imaging? Is the first sentence here trying to tell the reader that those patients with noncompliant ventricles (that did not change on imaging when their shunt failed and they had increased ICP) were excluded from this study? If yes this should be stated more clearly.
- If all patients had preop cross sectional brain imaging: Was there any correlation between the presence of transependymal edema and papilledema?
- You make the following strong statement in your Conclusion (lines 197-198): "Papillary oedema must be interpreted as a definitive sign of ICP elevation." This statement is actually not supported by your data. It is not as though you had a huge group of patients with confirmed papilledema and proved that all patients had ICP values in excess of 20 cmH2O. Rather, your study was structured the other way around and only included 8 patients with papilledema who had been preselected on the basis of having confirmed ICP > 20 cmH2O. So, personally, while I don't actually disagree with this statement, I also don't think this statement is appropriate for the Conclusion section of this particular paper given the data presented in the paper. I would not mind seeing this sort of statement in your Discussion section with supporting references and possibly a brief description of pseudopapilledema (as a statement this definitive should probably include acknowledgement of the fact that there are papilledema look-alikes that, while known to ophthalmologists, may not be common knowledge to all readers of this journal).
There are lots of minor grammar notes, probably needs review from a native English speaker prior to resubmission. Here are a few:
- "oppresses" in line 40 should probably be "compresses"
- The terminology “optical disc” is used in the paper a lot. I think the typical nomenclature of “optic disc” would be better. To my knowledge the terminology “optical disc” describes a flat, usually circular disc that encodes information (either analog or digital) in the form of pits and lands on a special material, often aluminum, on one of its flat surfaces. So, a compact disc, DVD, or Blu-Ray is disc is an optical disc, whereas this paper is discussing optic discs.
- Lines 51-52: Intracranial hypertension may be defined by pressures in excess of certain threshold values but it is not caused to "develop" because pressures are in excess of set threshold values. In other words, excessive pressures are not causative but rather diagnostic for intracranial hypertension.
- I'm terrible at explaining English grammar, but I know when something doesn't sound right and I'm sure "severely pallor" (line 104) doesn't sound right. Rather than "A severely pallor optical disc," it would be more grammatically correct to say "An optic disc with severe pallor."
- Lines 169-170: The first sentence of this paragraph does not make grammatical sense. The sentence ends with "this may be a helpful marker" but "this" is not defined earlier in the sentence. Assuming by "this" you mean papilledema it is still unclear what you're driving at here: If the whole point of your paper is that papilledema is not a reliable sign of elevated ICP then it is certainly not a reliable sign of a brain tumor (as not all tumors cause hydrocephalus/elevated ICP). So, even if the grammar was corrected I am still struggling to understand the point that you're trying make with this sentence.
- Why do you keep switching between "papillary oedema" and "papilledema"? - I would recommend sticking with "papilledema"
Author Response
Reviewer 2
My biggest concern with this manuscript is the lack of ICP data. If all patients had ICP measured why are these ICP measurements not included? I think it would be much more impactful data if these ICP measurements were included. Moreover, the methods would imply that any patient with a measured ICP > 20 cmH2O and a preop fundoscopic exam would be included but there is no mention of the ICP being sustained > 20 cmH2O for 5 minutes or more of monitoring. (We all know that persons without hydrocephalus can generate transient pressures in excess of 50 cmH2O with a forced Valsalva so some mention of the ICP being sustained > 20 cmH2O for 5 min or more for those patients with preoperative EVDs/ICP monitors is necessary. (I do think it is reasonable to infer that those shunt patients who had ICP measured under anesthesia at time of shunt revision were living at a pressure close to the pressure that they had measured in the OR for the duration of the their shunt malfunction symptoms as I am sure 5+ minutes of pressure recordings were not obtained for these patients. But, for this unique patient population in the study it would be valuable to know how far their measured ICP in the OR was from their baseline ICP or shunt valve pressure setting. We all know that ICP will go up with a shunt malfunction in a shunt dependent patient but how high it goes, or how much the pressure deviates from the patient's baseline pressure with a working shunt, can be quite different from patient to patient.) In the conclusion (line 142) it is mentioned that this paper is correlating ICP and papilledema but, without ICP measurements included I don't feel that this paper is in fact correlating these measurements (as correlating these measurements would, in my mind, necessitate providing both measured ICP values and papilledema grades 1-5 for each patient included in analysis). While not as critical as providing the ICP measurements, grading the papilledema observed would have been nice.
General comments:
- Paragraphs 3-5 (lines 44-57) seem unnecessary to the focus of the paper. While autoregulation, ICP changes with the respiratory cycle, and the Monroe-Kellie doctrine are are generally related to the topic of increased ICP these topic do not specifically have bearing on the focus of this paper: the relationship between ICP and the observation of papilledema.
Thank you for this comment. We have rewritten these paragraphs to be shorter, but we believe that it is essential to put forward these topics in the introduction of the paper.
- Why discuss pressure in torr in line 51 of the intro when it is discussed in cm H2O in the remainder of the paper?
We have fixed this error; thank you for pointing this out.
- The last paragraph of the intro (lines 82-84) feels like a bit of a non sequitur. Maybe something that can be saved for a discussion section if you want to discuss CPPRH more. But, as the last statement in the intro, particularly as a stand alone paragraph, it doesn't make much sense.
We have rewritten part of the introduction; thank you.
- The discussion of IBM SPSS use and p valve of 0.05 (lines 114-116) should be included in the Methods section not Results
We have moved it to the appropriate section, thank you.
- You mention the significance of open versus closed fontanelles a lot but at no point in the paper do you tell the authors how many patients in your study had an open fontanelle. We can infer that at least one did (minimum age = 0.05) but given the median age of 4.8 I am guessing there were not too many patients with open fontanelles. This needs to be clearly stated if it is going to be such a focus in the Discussion section.
- While I agree that an open fontanelle is protective against fulminant global increases in ICP (as head enlargement and fontanelle bulging can help to prevent extreme increases in ICP), I not aware of data to support the statement in lines 164-165: "If we diagnose a brain tumor in a child who already has closed fontanelles, the obstruction of the ventricular system occurs more often." I have cared for many young patients with open fontanelles that develop tumor related obstructive hydrocephalus...while their hydrocephalus symptoms may be less acute/severe than slightly older children with closed fontanelles I don't know of data to support that young children are less prone to developing obstructive hydrocephalus. This would need references to support this statement.
We made this statement based on our experience with young patients with open fontanelles, we thank you for sharing your experience. We have rephrased the paragraph.
- You mention the potential for selection bias in the discussion but I think this needs even more attention. In my pediatric neurosurgery practice I would generally not request a preoperative fundoscopic examination in a patient with clear clinical/imaging evidence of hydrocephalus/increased ICP. I only ask for an urgent preoperative fundoscopic exam in patients that have questionable signs of increased ICP on exam/imaging. If this is the case at your institution then this would constitute a massive selection bias far greater than what is described in the discussion. If your institutional standard is to make an effort to obtain a preoperative fundoscopic exam on all patients with possible (or definite) increased ICP then this should be stated clearly as this will help to allay some concerns with respect to selection bias.
Thank you for this comment. We are trying to obtain a preoperative fundoscopic exam if the situation allows it; however, this is not possible for all eligible paediatric patients for several reasons. This is why we mention the severity of the selection bias in the last paragraph of the Discussion.
- Lines 106-110: These two sentences don't go well together given that the imaging studies don't provide ICP values. I understand that ICP values were also recorded but having these sentences follow each other without a more detailed description of the ICP value measurements is confusing. Moreover, I'm a bit confused - wouldn't the tumor patients have also undergone preoperative imaging prior to going to OR or having an EVD or ICP monitor placed? Why are the shunt patients being singled out with respect to the mention of preoperative imaging? Is the first sentence here trying to tell the reader that those patients with noncompliant ventricles (that did not change on imaging when their shunt failed and they had increased ICP) were excluded from this study? If yes this should be stated more clearly.
We have rewritten the mentioned lines to make the paragraph clearer.
- If all patients had preop cross sectional brain imaging: Was there any correlation between the presence of transependymal edema and papilledema?
Thank you for this interesting question. All the patients in our study had preoperational brain imaging and we have found no correlation between the presence of trans-ependymal oedema and papilledema.
- You make the following strong statement in your Conclusion (lines 197-198): "Papilledema must be interpreted as a definitive sign of ICP elevation." This statement is actually not supported by your data. It is not as though you had a huge group of patients with confirmed papilledema and proved that all patients had ICP values in excess of 20 cmH2O. Rather, your study was structured the other way around and only included 8 patients with papilledema who had been preselected on the basis ofhaving confirmed ICP > 20 cmH2O. So, personally, while I don't actually disagree with this statement, I also don't think this statement is appropriate for the Conclusion section of this particular paper given the data presented in the paper. I would not mind seeing this sort of statement in your Discussion section with supporting references and possibly a brief description of pseudopapilledema (as a statement this definitive should probably include acknowledgement of the fact that there are papilledema look-alikes that, while known to ophthalmologists, may not be common knowledge to all readers of this journal).
Thank you for this comment. We have amended this statement to correlate better with our study.
There are lots of minor grammar notes, probably needs review from a native English speaker prior to resubmission. Here are a few:
- "oppresses" in line 40 should probably be "compresses"
Thank you, we had the grammar reviewed by a native English speaker before resubmission.
- The terminology “optical disc” is used in the paper a lot. I think the typical nomenclature of “optic disc” would be better. To my knowledge the terminology “optical disc” describes a flat, usually circular disc that encodes information (either analog or digital) in the form of pits and lands on a special material, often aluminum, on one of its flat surfaces. So, a compact disc, DVD, or Blu-Ray is disc is an optical disc, whereas this paper is discussing optic discs.
Thank you, we had the grammar reviewed by a native English speaker before resubmission.
- Lines 51-52: Intracranial hypertension may be defined by pressures in excess of certain threshold values but it is not caused to "develop" because pressures are in excess of set threshold values. In other words, excessive pressures are not causative but rather diagnostic for intracranial hypertension.
Thank you for this comment; we have rewritten the paragraph, as noted above.
- I'm terrible at explaining English grammar, but I know when something doesn't sound right and I'm sure "severely pallor" (line 104) doesn't sound right. Rather than "A severely pallor optical disc," it would be more grammatically correct to say "An optic disc with severe pallor."
Thank you, we had the grammar reviewed by a native English speaker before resubmission.
- Lines 169-170: The first sentence of this paragraph does not make grammatical sense. The sentence ends with "this may be a helpful marker" but "this" is not defined earlier in the sentence. Assuming by "this" you mean papilledema it is still unclear what you're driving at here: If the whole point of your paper is that papilledema is not a reliable sign of elevated ICP then it is certainly not a reliable sign of a brain tumor (as not all tumors cause hydrocephalus/elevated ICP). So, even if the grammar was corrected I am still struggling to understand the point that you're trying make with this sentence.
Thank you for this comment. We have changed the wording of the sentence to clarify the message; itis essential to remember brain tumour in DDx of papilledema.
- Why do you keep switching between "papilledema" and "papilledema"? - I would recommend sticking with "papilledema"
We thank the reviewer for this comment; we have unified the term in the article.
Round 2
Reviewer 1 Report
Thank you for the opportunity to re-review the manuscript. I have read the author’s responses to my comments. I can appreciate that they were not interested in pursuing additional statistical tests and respect their choice, however they may wish to rephrase their study aim so that their methods answer the question they pose (see below). My other two points have not yet been adequately addressed, namely how they chose their ICP cut-off and clarifying in the manuscript who evaluated the papilledema. It is true you mention examination by an ophthalmologist in lines 82-83, but this is part of your inclusion/exclusion criteria, not part of the methods for papilledema evaluation in the study, which is in the next paragraph. Since the whole paper is about the finding of papilledema, the methods about how it was done need to be organized and extremely clear. I suggest having a separate paragraph for everything to do with your papilledema evaluation methods, rather than scatter it across paragraphs.
Here are specific examples of my previously raised concerns:
- Line 49 – there should be references cited for normal values, particularly how 20 cm of H2O was chosen, it is insufficient to say this is an institutional value, it needs to be validated (this is a concern I previously raised, and it has not been sufficiently addressed). Like all things, there is a distribution of values, did the authors decide that anything above the 95th centile of ICP is abnormal, for example? How was 20 chosen, this needs to be explicit. In the paper you cite, Lee et al 2017, they found “The mean ICP was 19.9 ± 10.0 cm H2O among those without papilledema and 33.3 ± 9.1 cm H2O among those with papilledema”; you need to justify using a cut-off that is KNOWN to be poorly associated with the finding of papilledema, and you need to contextualize your results with the ICP measurements found in other studies.
- Line 83 – “our study aimed to determine papilledema’s reliability as a marker of increased ICP” As mentioned in my initial comments, this implies the type of study where you compare the detection of papilledema to a gold standard such as ICP measurement, (ie, a 2x2 table where you have papilledma yes/no and ICP yes/no), however this study only looked at those with increased ICP and looked at the proportion with papilledema. It might be more clear to state it this way, rather than make it sound like you’re looking for sensitivity and specificity (which is implied by saying “reliability”). Something like, “our study aimed to determine the proportion of patients with increased ICP with concurrent papilledema” would be more representative of the study you conducted.
- 88 – as previously mentioned “same clinician” is not clear, it seems to suggest the neurosurgeon examined the fundi; it could also mean an emergency room doctor or a pharmacist. It would be more clear to say “same ophthalmologist” Even better would be to state their experience; “Fundus examination for all patients was performed by the same board certified ophthalmologist with 10 years experience”. A separate paragraph dedicated to exactly who, and how, and what, was done for papilledema evaluation would be valuable.
Additional concerns about organization (In my previous feedback I did not get into more minor edits because I had significant concerns about major methodology)
Line 108 is an example of something in the results, which should be in the methods (how headache was evaluated)
Line 111-113 why is it essential? Does your study have the statistical power to know if this is a real finding? This kind of interpretation is usually in the discussion, rather than the results.
Author Response
Thank you for the opportunity to re-review the manuscript. I have read the author’s responses to my comments. I can appreciate that they were not interested in pursuing additional statistical tests and respect their choice, however they may wish to rephrase their study aim so that their methods answer the question they pose (see below). My other two points have not yet been adequately addressed, namely how they chose their ICP cut-off and clarifying in the manuscript who evaluated the papilledema. It is true you mention examination by an ophthalmologist in lines 82-83, but this is part of your inclusion/exclusion criteria, not part of the methods for papilledema evaluation in the study, which is in the next paragraph. Since the whole paper is about the finding of papilledema, the methods about how it was done need to be organized and extremely clear. I suggest having a separate paragraph for everything to do with your papilledema evaluation methods, rather than scatter it across paragraphs.
We kindly thank you for re-reviewing our manuscript.
We are sorry to hear that some of the comments were not adequately addressed, however, we must point out that by judging by some of your comments and especially the Lines you address you must have not received the most recent revision from the editorial office, as some of the comments are not fully applicable or the topics mentioned have already been moved to a different segment of the article.
Comments applicable have been incorporated to the article.
Here are specific examples of my previously raised concerns:
- Line 49 – there should be references cited for normal values, particularly how 20 cm of H2O was chosen, it is insufficient to say this is an institutional value, it needs to be validated (this is a concern I previously raised, and it has not been sufficiently addressed). Like all things, there is a distribution of values, did the authors decide that anything above the 95th centile of ICP is abnormal, for example? How was 20 chosen, this needs to be explicit. In the paper you cite, Lee et al 2017, they found “The mean ICP was 19.9 ± 10.0 cm H2O among those without papilledema and 33.3 ± 9.1 cm H2O among those with papilledema”; you need to justify using a cut-off that is KNOWN to be poorly associated with the finding of papilledema, and you need to contextualize your results with the ICP measurements found in other studies.
There are many studies evaluating where to draw the line between normal and increased intracranial pressure in children, we have quoted one of the studies we have based our findings on.
While Lee does have the values your provided mentioned in his study, we are presenting our findings with 20 cmH2O set as a cut-off. While there are numerous studies that have a chosen a different ICP value, we believe this values chosen by us to be reasonable and comparable to many other studies evaluating intracranial hypertension in children.
- Line 83 – “our study aimed to determine papilledema’s reliability as a marker of increased ICP” As mentioned in my initial comments, this implies the type of study where you compare the detection of papilledema to a gold standard such as ICP measurement, (ie, a 2x2 table where you have papilledma yes/no and ICP yes/no), however this study only looked at those with increased ICP and looked at the proportion with papilledema. It might be more clear to state it this way, rather than make it sound like you’re looking for sensitivity and specificity (which is implied by saying “reliability”). Something like, “our study aimed to determine the proportion of patients with increased ICP with concurrent papilledema” would be more representative of the study you conducted.
Thank you for this suggestion. We have changed the sentence in question. Whilst your suggestion is true, and we have incorporated this in the paragraph, we also believe our study shows the reliability of ICP which has been included in previous revisions.
- 88 – as previously mentioned “same clinician” is not clear, it seems to suggest the neurosurgeon examined the fundi; it could also mean an emergency room doctor or a pharmacist. It would be more clear to say “same ophthalmologist” Even better would be to state their experience; “Fundus examination for all patients was performed by the same board certified ophthalmologist with 10 years experience”. A separate paragraph dedicated to exactly who, and how, and what, was done for papilledema evaluation would be valuable.
Thank you for this suggestion. We have further clarified who exactly performed the examination. However, we do think that this description is adequate for this manuscript, as adding a whole paragraph to the article could lead the reader away from this topic, as papilledema evaluation is quite standardized.
Additional concerns about organization (In my previous feedback I did not get into more minor edits because I had significant concerns about major methodology)
Line 108 is an example of something in the results, which should be in the methods (how headache was evaluated)
Line 111-113 why is it essential? Does your study have the statistical power to know if this is a real finding? This kind of interpretation is usually in the discussion, rather than the results.
Unfortunately, these comments are not applicable at this point, as explained in our first response.
Reviewer 2 Report
While the relatively minor comments made at the first review have been responded to and addressed the primary concern of this reviewer: The lack of ICP data (and to a lesser extent the lack of papilledema grading) were not commented on by the authors in their response and have not been included in the new version of the manuscript. It continues to strike me that if ICP values were obtained for every patient then these values should be included in the data presented, particularly given the small size of the study. ICP is not binary data (elevated or not) but continuous data and, it should be treated in this fashion whenever possible, particularly when trying to correlated ICP values with the presence or absence of observed papilledema. Given the lack of ICP data this relatively small study does not add much if anything to the current literature. It has long been understood that some patients with elevated ICP will not have papilledema on fundoscopic exam. The thesis of this paper seems to be that this is far far more common than is generally appreciated in the literature and by practicing physicians. But, without providing the reader with the recorded ICP values it is hard to draw any meaningful conclusions from this paper in my opinion. For example, one might wonder if some of the shunted patients had "low pressure hydrocephalus" such that they had symptoms and change in ventricle size when their shunt failed and allowed their baseline ICP of 6 to up to 20. In this patient group I would not generally expect papilledema (and wouldn't even ask for a fundoscopic exam before going to OR to fix the shunt). So, without providing the reader with ICP values that were reportedly recorded for all patients it weakens the paper very significantly.
Author Response
We thank you for this extensive comment. We have included a comprehensive table of the ICP values recorded according to the primary diagnosis (with respective additions to “Materials and Methods”, “Results”, “Discussions” and “Conclusion”.
Every patient included in this study had ICP values higher than 20cmH2O as this has been set as our cut-off value to be included in this study (Line 85) and thus the “low pressure hydrocephalus” is not applicable in our study. The average ICP value for patients with confirmed papilledema has been added to the paragraph “results”.
While we do agree that ICP is continuous data and, in our experience, it can continuously increase if the patient is not treated, this increase is usually gradual and so we have put down the average ICP value at the time of the fundoscopic exam.
Each respective value for every patient included in this study is available upon reasonable request, as we believe putting raw table data into the article would not add much for the reader nor increase the quality of the paper.